# A Parallelizable Task Offloading Model with Trajectory-Prediction for Mobile Edge Networks

**DOI:** 10.3390/e24101464

**Published:** 2022-10-14

**Authors:** Pu Han, Lin Han, Bo Yuan, Jeng-Shyang Pan, Jiandong Shang

**Affiliations:** 1School of Computer and Artificial Intelligence, Zhengzhou University, Zhengzhou 450001, China; 2National Supercomputing Center in Zhengzhou, Zhengzhou University, Zhengzhou 450000, China; 3Nanyang Institute of Technology, No.80, Changjiang Road, Nanyang 473000, China; 4School of Informatics, University of Leicester, Leicester LE1 7RH, UK; 5College of Computer Science and Engineering, Shandong University of Science and Technology, Qingdao 266590, China; 6Department of Information Management, Chaoyang University of Technology, Taichung 413310, Taiwan

**Keywords:** edge computing, mobile edge network, parallelization, task offloading, trajectory prediction

## Abstract

As an emerging computing model, edge computing greatly expands the collaboration capabilities of the servers. It makes full use of the available resources around the users to quickly complete the task request coming from the terminal devices. Task offloading is a common solution for improving the efficiency of task execution on edge networks. However, the peculiarities of the edge networks, especially the random access of mobile devices, brings unpredictable challenges to the task offloading in a mobile edge network. In this paper, we propose a trajectory prediction model for moving targets in edge networks without users’ historical paths which represents their habitual movement trajectory. We also put forward a mobility-aware parallelizable task offloading strategy based on a trajectory prediction model and parallel mechanisms of tasks. In our experiments, we compared the hit ratio of the prediction model, network bandwidth and task execution efficiency of the edge networks by using the EUA data set. Experimental results showed that our model is much better than random, non-position prediction parallel, non-parallel strategy-based position prediction. Where the task offloading hit rate is closed to the user’s moving speed, when the speed is less 12.96 m/s, the hit rate can reach more than 80%. Meanwhile, we we also find that the bandwidth occupancy is significantly related to the degree of task parallelism and the number of services running on servers in the network. The parallel strategy can boost network bandwidth utilization by more than eight times when compared to a non-parallel policy as the number of parallel activities grows.

## 1. Introduction

With the popularization and widespread promotion of handheld mobile devices in the Internet of Things (IoT), the mobile computing model has become one of the most important forms of computing under the current hybrid Internet communication model [1]. In the mobile network, people can submit various computing tasks at any time with the nomadic devices in their hands [2]. Unlike the traditional resource-centering cloud architecture model [3,4], mobile edge computing is a kind of distributed computing model centered on edge services [5]. It could resolve the problems, such as edge network delay [6], task overload [7], and network congestion [8], by maximizing the utilization of edge computing resources [9].

Due to the limitation of edge network resources in mobile edge network(MEN), how to optimize the offloading of services and the allocation of resources has always been a challenging problem [10]. Although the promotion of 5G technology and devices have greatly improved the data transmission and service delay of the MEN, the random access of network devices, the variety of available network resources and the users mobility may limit the advantages of MEN. Many researchers proposed that providing services to users in the nearest place can reduce the network delay and improve the Quality of Service (QoS) [7,11,12]. J.Xu et al. [13] proposed a service proximity caching strategy, which responds to the user’s service request on the nearest edge server, and shortens the waiting time for service requests. Another approaches split the multiple user’s tasks into many parts, and distribute them to the terminal devices, edge servers, and cloud to share computing resources of MEN [14,15,16].

The user task request and offloading scheme in the MEN is closely related to the access location of the user equipment (UE), thus it is becoming a trend to introduce user’s mobility into the task offloading strategy of edge computing services [17]. Because of the randomness and unpredictability of the user behavior in MEN [18], as shown in Figure 1, the demand for the resources provided by the network is not determined when the UEs pass through the area covered by the edge servers. It is the real-time changes in network resources caused by this random behavior that affect the performance of task load strategies. Nadembega et al. discovered that the movement of users in an edge network has a great impact on the user experience and QoS of network in [19]. They also proposed a method to improve the efficiency of data transmission for edge network tasks. Therefore, it is worth considering the behavioral factors jointly when exploring task offloading solutions in MEN.

Figure 1 represents some moving UEs request services from edge networks (ENs). If no policy intervention is added during the tasks assignment, the terminal users have to wait around for their expected results [20]. It can be improved by the effective utilization of edge devices to guess the user’s location before task offloading [21,22]. This kind of method, by introducing the trajectory prediction based on a prior data involving users’ previous location into the tasks offloading in MEN, indeed improved the effective utilization of resources near the user. However, it is challenging to obtain the user’s personal location information.

In this paper, we proposed an approach to distribute the tasks in MEN based on a non-prior trajectory prediction model (NPTP) which can enhance the predictive ability of task offloading decision. In other words, we construct a dynamic user-centering edge network (UCEN) based on user’s movement trajectory, and surmise the next position of the users just following their current information in the coverage area of MEN. Then, a tasks offloading model can automatically adapt to the topology changes of UCENs based on the user’s movement. In addition, the offloading model distributes the subs-tasks of one service to different computing devices for concurrent execution, and the simulation experiments of services paralleling showed that the model has obvious advantages in the task’s execution efficiency and equipment occupation rate. In this article, we analyze current trajectory prediction methods and tasks assignment strategies in MEN, and make some improvements in how to reduce the dependence of trajectory prediction on historical information and subs-tasks offloading among the parallelizable tasks. In short, our contributions are the following:

(1) A non-prior trajectory prediction model (NPTP).

NPTP is based on random walking model, which can forecast the user’s next position through their current location and the moving direction. It provides an important reference for edge servers to optimize service offloading.

(2) A dynamic construction mechanism for mobile edge networks.

According to the user’s trajectory, UEs construct a mobile cloud dynamically to serve themselves, which is a mobile user-centering edge network (MUCEN) architecture. The MUCEN would be constructed along the user’s movement path provisionally. Therefore, the available computing devices, prepared for the user during their movement, are dynamically adjusted in the domain of MUCEN.

(3) Parallelizable Task Offloading Strategy Based Position Prediction (PTOSBPP) in MEN.

Due to the concurrent and asynchronous execution of the subs-tasks, we proposed a service offloading strategy for MEN that can adapt to dynamic user’s location.

The rest of this paper is organized as follows: Section 2 introduces the related methods about task offloading in MEN. It illustrates that how NPTP model to guess the next position of the UEs, and formulate the cost of task offloading in Section 3. Then, the algorithms of NPTP and PTOSBPP in detail are described in Section 4. Finally, we show the simulation results and analyze the performance of the algorithms, and conclude this study in Section 5.

## 2. Related Works

Edge computing is a new computing model after cloud computing and fog computing [9,11]. Its purpose is to assuage the computing pressure of the cloud with the idle and scattered resources around the UEs. Task offloading is a well-established approach to improve the edge computing performance [1,23]. However, with the increasing of the portable devices in MEN, it makes the task unloading and resource apportionment more complicated. The deployment of MEN services is a challenging issue during the process of tracking user’s mobility [24]. In order to achieve a better QoS and Quality of Experience (QoE) in MEN, lots of task offloading strategies are proposed by many researchers who are studying how to distribute the limited network resources to end users in a rational way. The traditional optimization techniques for these distribution schemes can be broadly divided into two categories: decision optimization and architecture or devices optimization.

Decision optimization mainly analyzes the parameters, bandwidths, server speeds, available memory etc., evaluates the cost function, and then determines a resource allocation strategy [25]. For instace, Chen, Xu et al. [26] designed a novel D2D Crowd framework to assist in the sharing of edge network resources. The authors of [27] proposed an adaptive computation offloading approach for saving the power of portable computer. Wolski et al. [28] presented a framework for making computation offloading decisions based on the networks bandwidth. The methods, proposed in [29,30], tried to explore as many computing resources as possible around the base station (BS) to meet the computing needs of edge services. As mentioned in [1], discovering and using the potential resources as much as possible is theoretically conducive to task execution efficiency in MEN. In most cases, these solutions are restricted by the number of free resources in UCENs, and weak in adapting to real-time changes of the network topology caused by random accessing. When the edge resources are insufficient, the cloud servers still need to provide a large amount of computing resources to assist in completing the computing tasks. Optimizing the network service architecture also address the issue of deal distribution of edge services. The UEs are responeded from the MEN in a short time though a novel network device [31,32] or virtualization technologies [25]. Femtolet proposed by Mukherjee et al. [32] could offer communication and computation offloading facilities simultaneously at low power and low latency. As well as, Chun et al. [33] disgned an architecture to extend the edge server by cloud. There are lots of article studying the methods mentioned above. We summerize them in Table 1, and analyze the shortcomings among various approaches.

Arbitrary accessing computational resource is a relatively prevalent occurrence in MEN, which causes complications in task offloading. Thus, if the destination server is sensed in advance for a moving user, it is beneficial for task offloading. As predicting the target edge sever is related to the user’s spatio-temporal location, it is necessary to study the mobile behavior of users. There are large numbers of mobile devices with random access in the MEN. It is the way of connection that brings huge challenges to the task offloading.Currently, this issue has attracted much attention of authors in [9,14,19,21,22,41]. A deep learning method in [37] is used to predict the trajectory of mobile edge network users for the task distribution among the ESs. O.Tao et al. [24] used the Markov model to predict the user’s movement trajectory, and achieved the optimal goal of long-term service in the application and released of network resources. Aissioui et al. [42] also proposed a FMeC-based framework for automated driving, which addresses the increased cost of network service migration caused by vehicle location updates. The methods proposed in [38,39] speculated the the UEs’ next position based on the terminal devices’ track. In some special scenario, the service offloading decision is built on a Lagrange trajectory interpolation method [40]. These studies indicate that mobility of the users has a certain impact on the efficiency of task execution in MEN, and the prediction of the user’s movement trajectory is a suitable for improving task offloading performance. However, all of methods have a common feature that they needed historical data for position pridiction. We call them as prior displacement prediction model (PDPM). In Section 1, we have already stated that it’s difficult and even illegal to acquire the past privacy data. Thus, PDPM seems to have some limitations in practice.

To overcome the shortage of PDPM in historical data collection, we propose a non-prior trajectory prediction model to predict the path of moving users in this paper. In addition, considering the techniques of paralleling and fragmentation could expand the number of edge devices that participate in task offloading, and reduce the network latency, the methods in [15,43] show that parallelization is one of the effective and efficient ways to improve the offloading of edge tasks in MEN. Therefore, our experiment also import a sub-task offloading policy.

## 3. Problem Statement and System Model

In this section, we formulate the mobile user-centering edge network architecture, introduce the model of the NPTP, and then discuss the optimal cost calculation method of parallel unloading strategy.

### 3.1. Mobile User-Centering Edge Network Architecture

Generally, the terminal device with users are centered on the edge server which dominates the task assignment. All of the service request are triggered by them. The hardware resource, such as processor. storage etc.,can also participate in the scheduling and allocation of resources. However, these resources are monopolized by the user device for security.

Different from the traditional MEN, the user-centering mobile edge network is a dynamic network architecture along with the path of moving UEs. As shown in Figure 2, the wireless mobile devices carried by users are randomly connected to the edge network, that leads to the rebuilding of a new network topology. Thus, the edge server needs to constantly adjust the assignment of tasks.

As the user ’s location changes in ENs, so does the serving ES (Figure 2). That may result in available resources providing to users to be different at a certain moment. In mobile user-centric edge network, the resources available around the user can be adjusted timely to cope with random changes by the user’s location.

Given $U=\left\{U_{i}|i=1,2,3,\ldots ,M\right\}$ represents the set of users in the network, where *M* is the number of users; $Dev=\left\{Dev_{i}|i=1,2,3,\ldots ,D\right\}$ represents the set of devices in the network, *D* is the number of devices; $ES=\left\{ES_{i}|i=1,2,3,\ldots ,S\right\}$ represents the set of ES, where *S* is the number of edge servers in the network. The user-centered edge service network can be expressed as a triple (U,D,ES).

$EN=\left\{EN_{i}|i=1,2,3,\ldots ,EC\right\}$ represents the set of ENs, and EC is the number of ENs in the network. Each edge network can be represented by a quintuple $EN_{i}=\left(UQ_{i},RQ_{i},TQ_{i},RS\_cur_{i},RS\_max_{i}\right)$, where UQ represents the user queue requesting the computing devices in MEN, RQ represents the available resource queue, TQ represents the task request queue in the EN, RS_cur represents the current usage of edge network resources, and RS_max represents the maximum load capacity of the EN (including the upper limit of connection number, CPU occupancy rate, bandwidth, etc.). All sets above must obey the constraints in Equation (Equation 1).
(1)$$\begin{array}{c} \left\{\begin{array}{c} \sum_{i=1}^{S}\left|UQ_{i}\right|\leq M\\ \forall |RQ_{i}|\leq |Dev_{i}|,1\leq i\leq S\\ \sum_{i=1}^{S}RS\_cur_{i}\leq \sum_{i=1}^{S}RS\_max_{i} \end{array}\right. \end{array}$$

Since the number of resources and network performance varies over time, we set $EN_{k}\left(t\right)=\left(UQ_{i}\left(t\right),RQ_{i}\left(t\right),TQ_{i},RS\_cur_{i}\left(t\right),RS\_max_{i}\right)$ to represent the $EN_{k}$ at time *t*, where $RS\_max_{i}$ is a constant independent of time; $Dev_{i}^{k}\left(t\right)$ represents the set of devices in $EN_{k}$ at *t*.

### 3.2. User Trajectory Prediction Model

Compared with the conventional prediction method, the user trajectory prediction model proposed in this paper is a non-priori prediction solution. Just according to the current motion state of the user, we guess the possible position of the user in the next time slice. The user’s movement refers to the random walk model which can change the direction and speed of the users randomly.

Assuming that the speed of user movement varies with time, which is denoted by v(t) at time *t*. For every fixed time slice *T*, the edge server needs to update the location of the user. Then, the moving distance *d* of the user in a single *T* can be expressed as d=T∗v(t), which directly affects the selection of the target device for the current request task offloading.

When the *T* infinitely approaches a very small value, we regard that the UEs movement in a straight line with a constant direction and speed. Therefore, we can deduce a circular region with radius r according to Equation (Equation 2) (red notation in Figure 3), which is the user’s activity range in the next time period.

Equation (Equation 2) represents that the UEs move in a straight line with a constant direction and speed, when the ε is a small value. Therefore, we can deduce a circular region with radius r according to Equation (Equation 2) (red notation in Figure 3), which is the user’s activity range in the next time period.
(2)$$r\left(t\right)=\lim_{T\to \varepsilon }T*v\left(t\right),$$
where ε is extremely small values, and *T* is the time interval for system polling.

According to the radius r determined by Equation (Equation 2), there is an unambiguous circular region for the moving UEs.In Figure 3, α, β, γ and θ are the vector angles between the velocity vector *v* and different adjacent communication ESs, which represent the range of service areas that the user is about to reach for different base stations. These angles divide the entire circular region into several fan-shaped regions. We choose the sector area corresponding to the smallest two angles as the range with the greatest probability of user’s next location, which is the region corresponding to ∠α and ∠β in Figure 3. Therefore, the position of the moving user at next time is most likely to fall within the domain delimited by Equation (Equation 3).
(3)$$\begin{array}{c} Area=Area(\beta ,r)+Area(\alpha ,r)\\ \;where\;\left\{\begin{array}{c} Area\left(\beta ,r\right)=\pi /360*\beta *r^{2}\\ Area\left(\alpha ,r\right)=\pi /360*\alpha *r^{2} \end{array}\right. \end{array}$$

The user activity range prediction model described in Figure 3 can outline the next location based on the current user’s status, which is an important basis for the target device selection when the subsequent terminal device task is unloaded. This model can effectively narrow down the search scope for available devices in the offloading algorithm.

### 3.3. Parallelizable Task Offloading Cost Model

In Figure 3, the candidate ES is determined by the size of the overlap between the sector area and the coverage area of the BS. The resources of current EN and alternative ES will participate in the offloading of the network task. In this section, it mainly introduces the offloading cost evaluation model in parallelizable task offloading strategy.

$BS=\left\{BS_{j}|j=1,2,3,\ldots ,BN\right\}$ represents the set of edge server BSs, BN is the number of BSs, $Task=\left\{task_{i}|i=1,2,3,\ldots ,N\right\}$ represents the set of tasks in the MEN, and *N* is the total number of tasks that can be requested in the network. Furthermore, each task has a parallelizable subtask sequence $sub\_task_{i}^{k}$, which is represented by a *k*-tuple $task_{i}=\left\{sub\_task_{i}^{1},sub\_task_{i}^{2},\ldots ,sub\_task_{i}^{k}\right\}$

For simplicity, each BS is represented by a quadruple $BS_{j}=\left\{EN,CU,BW,TQ\right\}$, where EN is the edge network to which the BS belongs, it contains the available resources in the network; CU represents CPU occupancy; BW represents the bandwidth usage of edge servers; TQ represents the task queue on the BS. Assuming that all BS servers in MEN possess same performance, so that CPU_MAX, Bandwidth_MAX and BS_Task_MAX respectively represent the upper limit of the server ’s computing power, bandwidth, and the number of tasks carried.

Each sub-task is represented by $sub\_task_{i}^{k}=\left(CPUi^{k},Datai^{k},Tolerance\_maxi^{k}\right)$, where CPU represents the number of CPU clock cycles of task; Data is the amount of data generated by each task in the execution process; Tolerance_max is the maximum delay of execution time of each sub-task. In this paper, we mainly considers the consumption of the task in the CPU clock cycle on the computing device and the data exchange in the network to evaluate the offloading strategy.

#### 3.3.1. Energy Consumption Calculation Model of Task Offloading

All resources around moving UEs will be used as the targets to unload each task, and the feasibility of this solution has been demonstrated in [44,45,46]. In Equation (Equation 4), $\xi_{e},\xi_{t}$ are CPU clock cycle rated energy consumption on remote cloud servers (CS), edge servers (ES), and terminal devices (TD), $task_{i}$ can be executed in parallel with *K* sub-tasks which have been divided by the designers of the task in advance, where $K_{1}$, $K_{2}$ and $K_{3}$ represent the number of sub-tasks assigned on CS, ES and TD. According to the offloading strategy proposed in the article, the energy consumption required for the execution of $task_{i}$ is shown in Equation (Equation 4).
(4)$$\begin{array}{c} Energy\_CPU\_task_{i}=\xi_{c}*\sum_{\delta =1}^{K_{1}}CPU_{sub\_task_{i}^{\delta }}^{\delta }\\ +\xi_{e}*\sum_{\omega =1}^{K_{2}}CPU_{sub\_task_{i}^{\omega }}^{\omega }+\xi_{t}*\sum_{\chi =1}^{K_{3}}CPU_{sub\_task_{i}^{\chi }}^{\chi };\\ where\;K_{1}+K_{2}+K_{3}=\left|task_{i}\right| \end{array}$$

In Equation (Equation 4), $\xi_{c}*\sum_{\delta =1}^{K_{1}}CPU_{sub\_task_{i}^{\delta }}^{\delta }$, $\xi_{e}*\sum_{\omega =1}^{K_{2}}CPU_{sub\_task_{i}^{\omega }}^{\omega }$, $\xi_{t}*\sum_{\chi =1}^{K_{3}}CPU_{sub\_task_{i}^{\chi }}^{\chi }$ represent the energy consumption of CPU cycles of tasks in cloud, edge and end device respectively.

Generality, the edge network described in this paper is an random network with multi-task and multi-user access, the same task may be requested by multiple users at the same point in time. So, we assume that $cnt_{i}^{t}$ represents the total number of requests for task at *t* time, and then the CPU energy consumption for the entire network is shown in Equation (Equation 5).
(5)$$Energy\_CPU\left(t\right)=\sum_{i=1}^{\left|Task\right|}Energy\_CPU_{task_{i}}*cnt_{i}^{t}$$

#### 3.3.2. Network Delay Calculation Model

Network delay is another important indicator to measure the offloading strategy. It includes the execution time of the task itself, the synchronization waiting time between subtasks, and the time for data transmission. However, the transmission is the main reason for excessive network latency [11]. Equation (Equation 6) is the total data amount of $task_{i}$, where $sub_{D}^{k}$ is the data amount of all sub-tasks of $task_{i}$.
(6)$$D_{task_{i}}=\sum_{k=1}^{K}sub\_D_{task_{i}}^{k},\;where\;K=\left|task_{i}\right|$$

Commonly, the data in the network can be divided into upload data and download data, and $sub\_D_{task_{i})}^{k}$ in Equation (Equation 6) can be expressed in Equation (Equation 7).
(7)$$sub\_D_{task_{)}}^{k}=up\_D_{task_{i}}^{k}+down\_D_{task_{i}}^{k}$$

In Equation (Equation 7), $up\_D_{task_{i}}^{k}$ and $down\_D_{task_{i}}^{k}$ represent sub-task upload and download data; If the communication rate is represented by *B*, the execution time of $task_{i}$ in the network can be represented as:(8)$$\begin{array}{cc} T\_delay_{task_{i}} & =\sum_{k=1}^{|task_{i}|}(up\_D_{task_{i}}^{k}/B\\ & +down\_D_{task_{i}}^{k}/B)\\ & +sys\_delay_{task_{i}} \end{array}$$

In Equation (Equation 8), $sys\_delay_{task_{i}}$ is the time delay caused by EN itself. $cnt_{i}^{t}$ represents the number of requests for $task_{i}$, the network delay of the whole network at *t* can refer to Equation (Equation 9).
(9)$$\begin{array}{cc} T\_delay(t) & =\sum_{i=1}^{\left|Task\right|}T\_delay_{task_{i}}*cnt_{i}^{t}\\ \; & =\sum_{i=1}^{\left|Task\right|}(\sum_{k=1}^{|task_{i}|}(up\_D_{task_{i}}^{k}/B\\ \; & \;+down\_D_{task_{i}}^{k}/B+sys\_delay_{task_{i}}))*cnt_{i}^{t} \end{array}$$

### 3.4. Problem Statement

In this section, we study the impact of user mobility behavior on task offloading in mobile edge network, and propose a task offloading solution for parallel tasks based on the user trajectory prediction model. We transform this issue into the problem of finding an optimal set of devices with the least cost for task offloading in a mobile user-centering network, which is represented as multi-objective optimization function in Equation (Equation 10).
(10)$$Cost_{task_{i}}=T\_delay_{task_{i}}+Energy\_CPU_{task_{i}}$$
where $Cost_{task_{i}}$ is the cost function of offloading $task_{i}$. Assuming *A* is the set of all assignments of task at *t*, the objective function of offloading cost (Equation (Equation 10)) can be transformed into Equation (Equation 11) which is regarded as a function of *t* and *A*. Thus, the execution time of the task and the computational resources consumed become the optimization targets.
(11)Cost(A,t)=T_delay(A,t)+Energy_CPU(A,t)

To put it clearly, we need to convert the Energy_CPU(A,t) into computer clock cycles which is measured in time. Therefore, the units of T_delay(A,t) and Energy_CPU(A,t) are then unified. The assumptions and conversion rules are listed in the experimental section. Considering the parallelization of tasks, the each terms in Equation (Equation 11) are non-negative. The minimum of offloading cost is obtained when both of $T\_delay_{task_{i}}$ and $Energy\_CPU_{task_{i}}$ reach minimum values. Therefore, the process of finding the optimal solution of Cost(A,t) can be converted to two minimization functions (Equation (Equation 12)).
(12)$$\left\{\begin{array}{c} Min\left\{T\_delay(A,t)\right\}\\ Min\left\{Energy\_CPU(A,t)\right\} \end{array}\right.$$

Based on the constraint of Equation (Equation 12), we try to seek an optimal *A* all tasks at *t*. However, it is hard to search a solution of Cost(A,t) to satisfy both of the conditions in Equation (Equation 12). We assume that A is the subset of *A* which makes the T_delay(A,t), and B is the subset of *A* which makes the Energy_CPU(A,t). Therefore, the optimal solution Equation (Equation 11) is divided into two cases. If A⋂B≠Φ, there is a optimal solution for Cost(A,t). Otherwise, Cost(A,t) only has a suboptimal solution.

## 4. Mobility-Aware Parallelizable Task Offloading Strategy (MPTOS)

Based on the models discussed in the previous section, this section describes the process of NPTP and PTOSBPP in detail. A mobility-aware parallelizable task offloading strategy is proposed here, the task offloading decision is based on the NPTP.

### 4.1. Overview of the MPTOS

As the UEs shuttle in the EN, the task requesting of moving UEs is irregular. The ESs perceive the user’s location according to task application from the end devices and passively provide service for them. In order to solve the NP-hard problem [30,47,48], task unloading strategy must be able to adapt to the changes of EN caused by the user’s movement. It is MPTOS that can dynamically modify the offloading policy as the ENs topology changes. MPTOS also includes user trajectory prediction algorithm without prior information and task offloading algorithm for parallelizable task. To demonstrate more clearly the task offloading of MPTOS, we present a complete description of the task offloading and decision-making process in Algorithms 1 and 2 in this section.

### 4.2. Non-Prior Trajectory Prediction Algorithm

Non-prior information trajectory prediction algorithm is the basis for the implementation of following service offloading strategy. The algorithm predicts the location of the UEs in the next slice by the instantaneous position and current speed of the UEs. Then, PTOSBPP unloads the request service received by the current device to the target edge device in advance according to the feedback results of NPTPat *t*, which is convenient for terminal user service requests at t+1.

According to the model in Figure 3, it quantifies the position angle between the motion vector $\vec{v}$ and the edge servers, which determines the next hop ES. Then, the offloading tactics works on the current available device queue including the next target ES. The algorithm of NPTPis shown in Algorithm 1.
**Algorithm 1** Non-prior Trajectory Prediction algorithm (NPTP)**Require:**  cu_dev_coordinate: Current coordinate of the user’s device  cur_BS_coordinate: Current coordinate of the servicing BS  speed: Current user’s speed that includes a direction and a value of mobile rate  R: Radius of BS covered area**Ensure:**  Next_BS: Candidate device for the task offloading  1:   Mobile_r ←$\left|speed\right|*T$
  2:   Neighbor_BS ← get_neighborBS()  3:   dis ← distance(cur_dev_coordinate, cur_BS_coordinate)  4:   Mobile_area ← pi * Mobile_r * Mobile_r  5:   **if**
dis<=R˘Mobile_r
**then**
  6:  Next_BS ← cur_BS_coordinate  7:   **else**  8:  //Get the angle between the velocity vector and different adjacent BS, and save the    result in Neighbor_ angle  9:  **for**
i∈(0,len(Neighbor_BS)
**do**
10:    temp_v ← make_vector(Neighbor_ BS [i], cur_dev_coordinate)11:    Neighbor_ angle[i] ← get_angle(speed,temp_v)12:  **end for**13:  min_angle ← Min(Neighbor_ angle)14:  Next_BS ← **getBS_from_angle(minangle)**15:   **end if**16:   retrun Next_BS


### 4.3. Parallelizable Task Offloading Strategy Based Position Prediction

In order to adapt to the movement of UEs, the algorithm of PTOSBPP, combined with the characteristics of parallel task in ENs, has to select the optimal unloading target device from available device queues in each time slice according to Equation (Equation 12) in Section 3.4. Thus, the strategy of PTOSBPP conform the following principles to distribute the tasks in ENs:

(1) MEN does its best to satisfy users’ resource requests.In addition, the time dependence of edge task on UEs’ displacement should be considered comprehensively in the provision of services by MEN.

(2) Large tasks that take up more CPU cycles are preferentially distributed to the devices with better bandwidth and sufficient-resources ESs. That can shorten the execution time of a single task.

(3) The number of tasks offloaded on ESs is restricted by increasing the parallelization of subtasks. In other words, when network delay caused by mobility cannot be avoided, cloud server and end device collaboration ought to be utilized to its full potentials.

Algorithm 2 explains the scheme of PTOSBPPin detail. The US sends different types of task requests to different edge servers by the user’s trajectory at each time slice, the PTOSBPP will be running on each ESs to offload the tasks to different candidate equipment.
**Algorithm 2** Parallelizable Task Offloading Strategy Based Position Prediction (PTOSBPP)**Require:**  M,N: Number of users and subtask of per $task_{i}$  subtask[M][N]: Subtasks list of $task_{i}$  cur_dev,cur_BS,cloud: Current user’s device, servicing BS and cloud servers  pre_BS[M]: Candidate BSs for M users selected in the Algorithm 1**Ensure:**  A[M][N]: Device assignment list of each subtask contained in subtask[M][N]  1:   Initialization of the information of UEs, edge servers, cloud and tasks  2:   Make array of subtask randomly  3:   **for**
i∈(0, M ) **do**  4:  $k\in \left\{et|exe\_Time(subtask[i][et])>T\right\}$  5:  $t\in \left\{et|exe\_Time(subtask[i][et])\leq T\right\}$  6:  **if** pre_BS[i] == cur_BS **then**  7:    **if** len(cur_dev.TQ) ≤ MAX **then**  8:     A[i][t] ← cur_dev  9:    **else**10:     A[i][t] ← cur_BS11:    **end if**12:    **if** len(cur_BS.TQ) ≤ BS_Task_MAX **then**13:     A[i][k] ← cur_BS14:    **else**15:     A[i][k] ← cloud16:    **end if**17:  **end if**18:  **for**
j∈(0,N)
**do**19:    **if** len(cur_BS.TQ)≤ BS_Task_MAX **then**20:     A[i][t] ← cur_BS21:    **else**22:     A[i][t] ← cur_dev23:    **end if**24:    **if** len(pre_BS[i].RQ) ≤ BS_Task_MAX **then**25:     A[i][k] ← pre_BS26:    **else**27:     A[i][k] ← cloud28:    **end if**29:  **end for**30:   **end for**31:   return A

## 5. Experiment and Discussion

In this section, we evaluate the proposed approaches by a series of experiments. All the experiments are conducted on a Windows OS equipped with Intel Core i5-8500 and 16 GB RAM. All of the algorithms are completed with python3.7 and pycharm IDE. We take a part of the EUA data set in these experiments, which contains the location information of 125 base station sites and 816 users in a Central Business District(CBD). EUA data set is a publicly released to facilitate research in Edge Computing, we can get it from the internet freely (https://www.opensourceagenda.com/projects/eua-dataset (accessed on 12 October 2022)). In addition, we also construct and use a task data set with 8 types, and each task is divided into several sub-tasks. According to the random walk model in Section 3, the users move in the area of a Central Business District, and send different task requests to the edge server in time slices stochastically.

### 5.1. Benchmark Approaches

As there is no completely similar experimental scenario to the issue discussed in this paper, we abstract three experimental schemes from the previous literature [9,14,19,21,24,35] to compare the performance of our methods. The tasks offloading method, proposed in this paper, is based on user trajectory prediction and introduces the parallel execution strategy of sub-tasks. By combining the random walk model with the EUA data set, we have designed three experimental scenarios to verify the effectiveness of our method. The following benchmark experiments are different combinations of NPTP and parallelizable task offloading strategy, and the detail are shown in Table 2.

(1) Random strategy (RS)

RS randomly assigns the undivided task to the devices surrounding the UEs. In RS, since the tasks are not split into multiple sub-tasks, the UEs, ESs and cloud servers cannot execute in parallel. If a task can be completed in one time slice, it will be unloaded to the neighbor ES. Otherwise, the UE requesting the task is responsible for completing the task.

(2) Non-position Prediction Parallel Strategy (NPPS)

In non-position prediction parallel strategy, the task is split into several parallel sub-tasks. That is the same task at the same time that can be divided into multiple sub-tasks and run in parallel between the UEs, ESs and cloud servers. This strategy may improve the execution efficiency of tasks within a single time slice. However, in order to avoid repeated requests during UEs moving and the failure to feedback the results to the terminal devices, the sub-tasks whose execution time exceeds one time slice are unloaded to the UEs for execution firstly.

(3) Non-parallel Strategy Based Position Prediction (NPSBPP)

In NPSBPP, it introduces the user trajectory prediction model mentioned in Section 3, and the edge resources predicted by model are also considered as the candidate devices for unloading the current task. Meanwhile, the task that runs more than one time slice is offloaded beforehand to the cloud or the ES at the next slice.

### 5.2. Experimental Setting and Evaluation Standard

To improve the universality of our approaches, our experiment Section 5 groups of fixed number of users from 816 users in the EUA data set. These users move at different speed based on the random walk model proposed in Section 3, and request for different types of tasks randomly at each time slice. The simulation experiment is carried out under four different task offloading strategies. In order to ensure the consistency of the experiment, the parameters in the simulation environment are set as follows in Table 3:

To comprehensively analyze the performance of our approaches, we choose a series of comparison indicators to evaluate our method, such as task offloading predicting hit rate, network bandwidth, and task execution efficiency. Where task offloading predicting hit rate is equal to the accuracy rate of user’s position prediction, bandwidth is mainly used to measure the burden of task parallel strategy, and the task execution efficiency is quantified the task acceleration ratio between parallel mode and serial mode.

### 5.3. Results and Discussion

In this section, we display the experimental results from the predicted hit rate of the mobile users, bandwidth, and task execution efficiency and compare them with RS, NPPS and NPSBPP.

#### 5.3.1. Task Offloading Hit Rate

Among the four strategies, NPPS and PTOSBPP adopt the position prediction method proposed in Section 3 to offloading the tasks, so that the task offloading hit rate and its comparison will be obtained from these two experiments. In each experimental scenario, we simulate the users’ trajectory by the random walk model, and collect the track information of 400 randomly selected users in 5 groups. According to prediction model in Figure 3, we compare the location of each user at time slice of *t* and t+1 by counting the number of times that the front and back positions are the same. Furthermore, we make statistics of these data according to 4 ranges of speed to observe the impact of rate on the prediction accuracy. Figure 4a,b present the predicting hit rate of NPPS and PTOSBPPamong different user sets. Since no prediction strategy is used in RS and NPPS, there is no need to compare this item in these two cases.

Each line of Figure 4 displays the hit ratio trend for different user data sets. As shown in Figure 4, with the increase of user’s moving speed, the accuracy rate gradually decreases. However, they still remain above 75%. In other words, when there is no sufficient additional information, the faster the user moves, the harder it is to predict the user location.

#### 5.3.2. Bandwidth

Any task unloading solution, especially collaboration strategy of UEs, ESs and clouds, has a certain impact on the EN. NPPS and PTOSBPPare offloading strategies for parallel tasks. In the simulation experiments, we have calculated the bandwidth occupancy of different offloading strategies at each time. Since more sub-tasks in the parallel offloading strategy may be assigned to different edge devices for execution shown in Figure 5a, it can be seen from Figure 5b that the average bandwidth utilization of NPSBPP and PTOSBPPis higher than RS and NPPS. Nevertheless, all of the curves in Figure 5b do not reach half of the total bandwidth.

#### 5.3.3. Execution Efficiency of Task

Eight different types of task requests are randomly generated at each time slice in the simulation experiments. Each task can be divided into several sub-tasks. Under different strategies, we collected the task execution time of 400 random users, calculated the average ratio of the actual execution time to the theoretical time of all users’ requesting tasks. Figure 6a presents the average number of tasks or sub-tasks that needs to be unloaded in MEN at each point in time, and the Figure 6b shows that the average ratio of the actual task execution time to the theoretical practice time under different unloading scheme whose horizontal axis is the UEs’ moving speed. The actual task execution time of RS and NPPS is over four times more than theoretical value, while the NPSBPP and PTOSBPPare basically close to the theoretical time.

As can be seen from Figure 6, the execution efficiency of programs in PTOSBPP and NPSBPP have obvious advantages over RS and NPPS. Since the parallel mechanism, the actual execution time of all tasks is close to the theoretical time in PTOSBPP, while the execution efficiency of NPSBPP is slightly higher than that of PTOSBPP. These results suggest that the prediction of UEs’ position in task offloading can effectively improve the execution efficiency of tasks in MEN, and the parallel division of tasks can shorten the execution time of tasks as well.

## 6. Conclusions and Future Work

In this paper, we identified the importance of prediction of user movement trajectory in task offloading of MEN network. We proposed a task offloading strategy for paralleling edge network based on a prediction model of non-prioritized trajectory. The time complexity of two algorithms has certain advantages to some extent. For example, the time complexity of Algorithm 1 seems to be O(n), but the *n* is the number of neighbor edge servers which is not more excessive. Meanwhile, the time complexity of Algorithm 2 is O(M×N), where *M* and *N* are the number of UEs and subtask. As the data structure of device assignment list, the temporal consumption of this algorithm needs to improve in the future.

Experimental results show that the prediction of user location not only improves the hit rate of target edge server for task offloading in parallel networks, but also effectively improves the execution efficiency of tasks and the utilization rate of network resources. Meanwhile, it demonstrated that the user mobility rate is the key factor affecting the above performance. However, we found that the hit rate of task offloading prediction decreased as the user moved faster. Therefore, how to improve the hit rate of the task offloading in fast moving scenarios is the one of our research challenges and key points in the future.

Additionally, mobile user-centered architecture is a type of dynamic networking approach in MEN, it is appropriate for the accessing way of MEN. In these networks, it may provide a personalized list of available devices for any end user, and increase the equipment utilization of the edge network. Since the user device participates in managing the resource of edge networks, the security of the network is a inevitable issue, which also is a potential area for research in our next works.

## Figures and Tables

**Figure 1 entropy-24-01464-f001:**
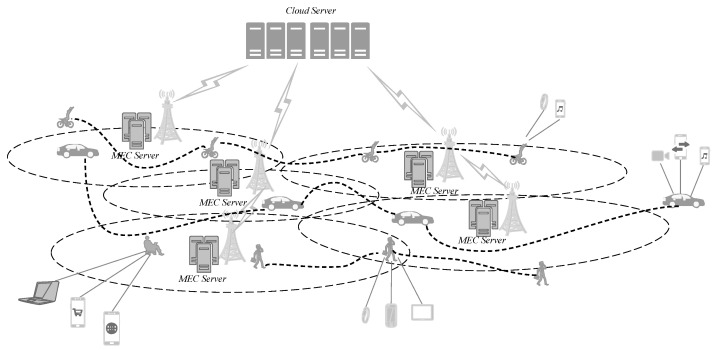
Moving targets in the MEN.

**Figure 2 entropy-24-01464-f002:**
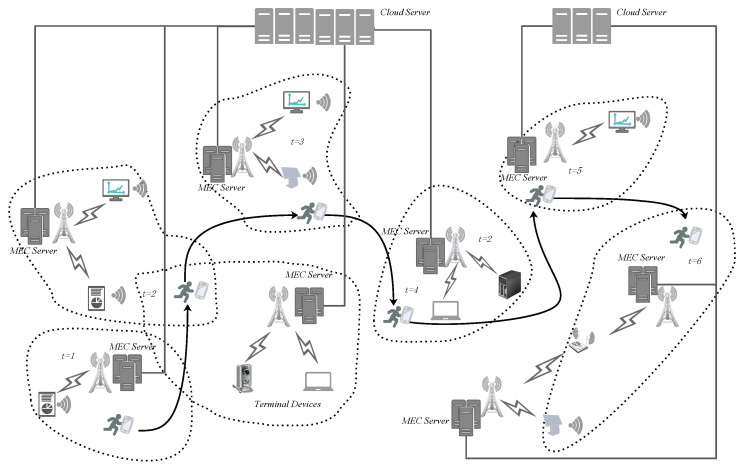
Architecture of User-centered edge mobile network.

**Figure 3 entropy-24-01464-f003:**
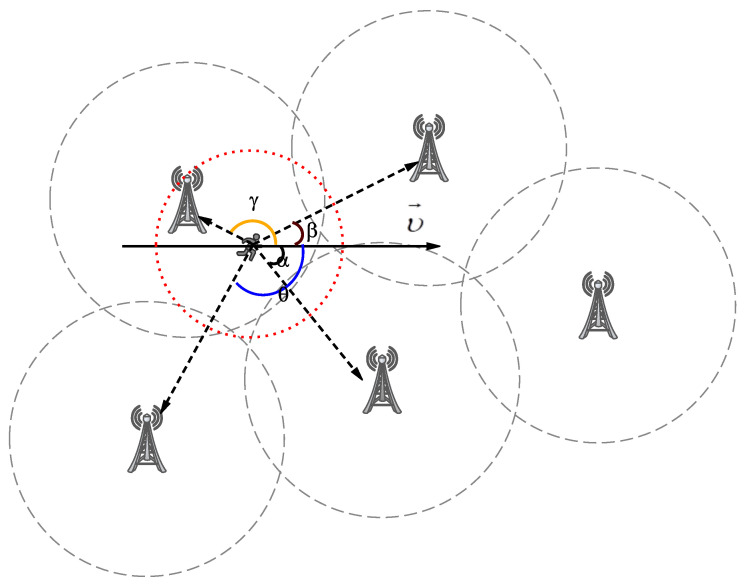
Prediction Model of Mobile user trajectory.

**Figure 4 entropy-24-01464-f004:**
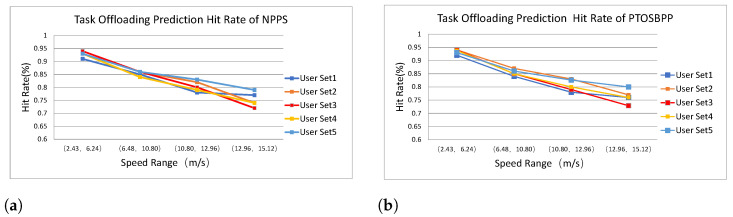
Task Offloading Hit Rate of NPPS and PTOSBPP. (**a**) Task Offloading Hit Rate of NPPS; (**b**) Task Offloading Hit Rate of PTOSBPP.

**Figure 5 entropy-24-01464-f005:**
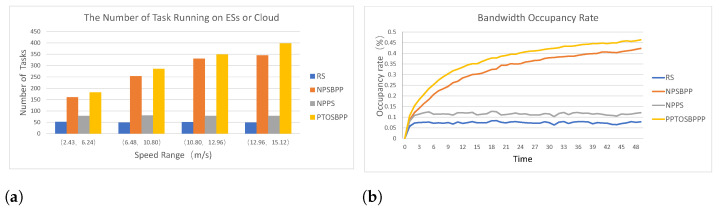
Bandwidth occupancy rate of different strategies. (**a**) The Number of Tasks Running on ESs or Cloud; (**b**) Bandwidth Occupancy Rate at Different Time.

**Figure 6 entropy-24-01464-f006:**
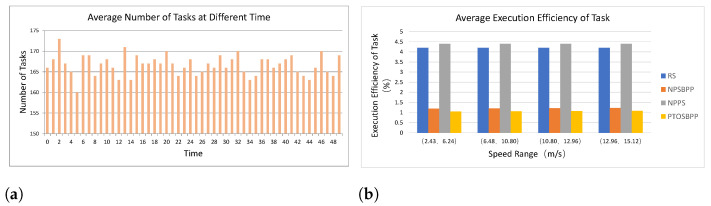
Average Execution Efficiency of Task. (**a**) Task Requests at Different Time Slice; (**b**) Average Execution Efficiency of Tasks among Different Cases.

**Table 1 entropy-24-01464-t001:** Categories of Traditional Task Offloading Research in Edge Networks.

Optimization Goals		Methodology Description	References	Shortcoming
Decision	Offloading based on constant resources	Assigning computational tasks to user’s neighbouring devices based on the resources in the current area	[1,4,13,16,23,26,27,28,29,30,34]	The offloading stratagy is limited by the number of available resources.
	Offloading based on extended resources	Expanding the available resources set based on the user’s current area and the predicted target area, and then offloading the tasks	[19,21,22,35,36,37,38,39,40]	Predictive modeling needs historical movement trajectory, it’s difficult to obtain these personal data.
Architecture or Devices		Optimizing the architecture of edge network or update devices to improve the performance of task offloading	[3,5,18,20,31,32,33]	It requires additional investment to upgrade and renovate the network.

**Table 2 entropy-24-01464-t002:** Detail of Different Experiments.

Experimental Scheme	Position Prediction	Parallel Strategy
Random strategy (RS)	✕	✕
Non-position Prediction Parallel Strategy (NPPS)	✕	✓
Non-parallel Strategy Based Position Prediction (NPSBPP)	✓	✕
Parallelizable Task Offloading Strategy Based Position Prediction (PTOSBPP)	✓	✓

**Table 3 entropy-24-01464-t003:** Experimental Parameters Setting.

Object	Parameter Setting	
End-users	Select 80 users as a group randomly from 816 users in EUA dataset, that might ensure the users are distributed within the CDB randomly; each experimental scenario contains 5 groups users.	
Task	8 different types of task, each task is represented by a subtask set $\left\{subtask_{i}|subtask_{i}=\left(CPU,Data,Torlarence\_max\right),0\leq i\leq N\right\}$, where *N* is the total number of sub-tasks. We also assume that each task contains 2 to 5 subtask.	
Cloud/MEN Servers	There are 125 edge servers and one cloud server in the CBD area, and we assuming the resource of cloud is sufficient.	
Coverage Radius of BS (m)	150≤R≤400	
Bandwidth	We assume that the bandwidth between the end, the edge and the cloud is adequate. $BW_{end\_to\_edge}=100$ MB/s; $BW_{edge\_to\_cloud}=1000$ MB/s	
Time Slot	T is a constant, we set T=2s; and the number of task requests generated by each user in each time slice is in the range [0,3]	
Rated Cycle of CPU	The cycle of CPU of Cloud $\delta_{c}=10$	$\delta_{c}\gg \delta_{e}\gg \delta_{t}$
	The cycle of CPU of MEN servers $\delta_{e}=40$	
	The cycle of CPU of terminal devices $\delta_{t}=100$	
Moving speed (m/s)	$\vartheta_{1}:2\leq \vartheta_{1}\leq 6$	$\vartheta_{i}$ is the speed of different targets, and $\vartheta_{1}<\vartheta_{2}<\vartheta_{3}<\vartheta_{4}$
	$\vartheta_{2}:6\leq \vartheta_{2}\leq 10$	
	$\vartheta_{3}:10\leq \vartheta_{3}\leq 13$	
	$\vartheta_{4}:13\leq \vartheta \leq 16$	

## Data Availability

The data that support the findings of this study are available from the corresponding author upon reasonable request.

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
