# Peer review of "A Parallelizable Task Offloading Model with Trajectory-Prediction for Mobile Edge Networks"

_entropy, 2022, doi:10.3390/e24101464_

Round 1
Reviewer 1 Report
A Parallelizable Task Offloading Model with Trajectory-Prediction for Mobile Edge Networks
This paper proposes a dynamic user-centering edge network (UCEN) based on user’s movement trajectory and surmised the next position of the users just following their current information in the coverage area of MEC networks. And then, a tasks offloading model can automatically adapt to the topology changes of UCENs based on the user’s movement. In addition, the offloading model distributed the subs-tasks of one service to different computing devices for concurrent execution.
There are clear weaknesses in the paper that the authors must particularly pay attention and handle:
General comment:
· The abstract must summarize the performance evaluation results.
· The review for most of the cited work is nebulous. The review failed to summarize critical details about each of the cited research study and its relevant pros and cons. Further, there is no comparison with the work in this article.
· Figures need to be amended, where the font in figure is slightly small to be seen which make them difficult to read.
· More details about the simulation software exploited should be added.
· The list of references should be reformatted and checked again to be matched with the journal requirement where a different styles and types are used. Please check some spells and typos.
· A thorough proofreading is required (best by a native English speaker).
Specific comments:
· The authors don’t mention how the dependency between tasks is addressed.
· How the proposed model affects on the user experience and quality of service?
· In equation 11, the authors sum time and energy together, however each of them has different unit. Please, clarify how this can be done?
· Some equations require further elaborations, such as equation 2. Moreover, in equation 11, the authors sum time and energy together, however each of them has different unit. Please, clarify how this can be done? So, I strongly recommend that the authors re-visit all equations and make sure consistent explanation for each is available.
· The related doesn’t cover the literature and state-of-the-art of the offloading and coordination aspects of the presented work appropriately. Therefore, I recommend that the authors add more literature about this point.
· In Section 4.1, the authors claim that the formulated problem is an NP-hard problem. However, I cannot find a direct evidence/method to prove that problem is an NP-hard problem. A clear proof by reduction is required.
· The authors should analyze the time complexity of the proposed algorithms in detail.
· At end of section 5, authors must compare their results to selected close papers from section 2.
Author Response
Dear reviewer:
Thank you for your decision and constructive comments on my manuscript. We have carefully considered the suggestion of Reviewer and make some changes. We have tried our best to improve and made some changes in the manuscript. Please find my itemized responses in below and my revisions/corrections in the re-submitted files.
I have replied the 14 comments one by one in the following:
Point 1:
The abstract must summarize the performance evaluation results.
Response 1:
Thanks for your advice and reminder. I have revised the abstract and added the relevant context about the performance evaluation results. The following is the modified screenshot.
Point 2: The review for most of the cited work is nebulous. The review failed to summarize critical details about each of the cited research study and its relevant pros and cons. Further, there is no comparison with the work in this article.
Response 2:
Please forgive our mistakes. I do find these deficiencies you pointed out. I reorganize the first two sections of the manuscript, remove some literature with weakly relevant content, and update several references.
In section 2, we summarize traditional task offlaoding methods in the first two paragraphs, and analyze limitations of this type of method when the edge resources are insufficient in paragraph 3. Paragraph 4 introduces several task offloading approaches with prediction of user trajectory, meanwhile, we illustrate the limitations of these methods and the reasons for the limitations. At the end of this section, we briefly describe our method and how our method overcome the weaknesses of other approaches.
Point 3: Figures need to be amended, where the font in figure is slightly small to be seen which make them difficult to read.
Response 3:
Thanks for your suggestion, we have increased the font size of the text on the figure 2 and figure 10, and update the content on page 5 and page 20 respectively.
Point 4: More details about the simulation software exploited should be added.
Response 4:
I have updated the details about the data set in the experiment, and add the link address of EUA on page 10. The following is the modified screenshot.
Point 5: The list of references should be reformatted and checked again to be matched with the journal requirement where a different styles and types are used. Please check some spells and typos.
Response 5:
Thanks for your reminding. I have check the format of the references again, and rebuild a .bib file following the Google Scholar citation format.
Point 6: A thorough proofreading is required (best by a native English speaker).
Response 6:
We apologize for the poor language of our manuscript. We worked on the manuscript for a long time and the repeated addition and removal of sentences and sections obviously led to poor readability. We check the article again carefully and find some expression errors. We have modified the abstract, introduction, and related works etc. , and aslo revised some errors in grammar and presentation. For example:
Line 61:” constructed” -> ”construct”
Line 62: ” surmised” -> ” surmise”
Line 65: “…adapt to the topology changes of UCENs based on the user's movement.” -> “adapt to the topology changes of UCENs.”
Line 65: “distributed” -> “distributes”
Line 67: “showed” -> “show”
Line 69: “analyzed” -> “analyze”
Line 75: “… current location and moving direction” -> “… current location and the moving direction”
Line 78: “UEs can build…” -> “UEs construct…”
Line 81: “…the computing devices…” -> “…the available computing devices…”
Line 83: “A offloading strategy …” -> “An offloading strategy …”
Line 158: “we shall first formulate the mobile” -> “we formulate the mobile”
Line 166: “rebuilding of a new topology” -> “rebuilding of a new network topology”
Line 172: “, andMis the number…” -> “where M is the number…”
Line 174: “S is the number…” -> “where S is the number…”
Line 203: “…offloading algorithm in this paper.” -> “…d offloading algorithm paper.”
Line 230: “, edge and end.” -> “, edge and end respectively.”
Line 280: “moving UEs is a irregular.” -> “moving UEs is irregular.”
In addition, I corrected some punctuation errors directly in the original draft.
Point 7: The authors don’t mention how the dependency between tasks is addressed.
Response 7:
The test dataset of tasks is generated based on the dataset EUA, it contains users, tasks,base stations etc. We illustrate the generating rule of the raw data in table 3 on page 13. We select 80 users as a group randomly from 816 users in EUA dataset; each experimental scenario contains 5 groups users. We also set 8 different types of task, and each task map to a random number N which present the total number of concurrently executable subtasks.
Point 8: How the proposed model affects on the user experience and quality of service?
Response 8:
We are able to predict a target edge server with the proposed model. This computing device is the edge server most likely to be required by the terminal in the next time slice. If we offload the task to this server, it reduce the time comsuing of network routing and transmission during the backhaul of computation results.
Point 9: In equation 11, the authors sum time and energy together, however each of them has different unit. Please, clarify how this can be done?
Response 9:
In our experiments, we convert the into computer clock cycles which is measured in time. So that, the result of equation 11 is presented by time. The conversion relationships for the number of CPU clocks is listed in table 3.
Point 10: Some equations require further elaborations, such as equation 2. Moreover, in equation 11, the authors sum time and energy together, however each of them has different unit. Please, clarify how this can be done? So, I strongly recommend that the authors re-visit all equations and make sure consistent explanation for each is available.
Response 10:
Thanks for your reminding again. I lost the decalration of symbol in equation 2. The following is modified screenshot of equation 2.
To avoid misunderstandings of equation 11 among readers, I add the illustration under the equation 11.
In addtion, I also recheck the each equation in the paper, and did some tiny corrections.
Point 11: The related doesn’t cover the literature and state-of-the-art of the offloading and coordination aspects of the presented work appropriately. Therefore, I recommend that the authors add more literature about this point.
Response 11:
I revised this mistake according to you comment 2. I also remove some literatures with weakly relevant content, and replace them by another references listed in the following:
[2] Shakarami, A.; Ghobaei-Arani, M.; Shahidinejad, A. A survey on the computation offloading approaches in mobile edge computing: A machine learning-based perspective. Computer Networks 2020, 182, 107496.
[26] Kumar, K.; Liu, J.; Lu, Y.H.; Bhargava, B. A survey of computation offloading for mobile systems. Mobile networks and Applications 2013, 18, 129–140.
[27] Chen, X.; Pu, L.; Gao, L.;Wu,W.;Wu, D. Exploiting massive D2D collaboration for energy-efficient mobile edge computing. IEEE Wireless communications 2017, 24, 64–71.
[31] Hirsch, M.; Mateos, C.; Zunino, A. Augmenting computing capabilities at the edge by jointly exploiting mobile devices: A survey.Future Generation Computer Systems 2018, 88, 644–662.
[32] Mukherjee, A.; De, D. Femtolet: A novel fifth generation network device for green mobile cloud computing. Simulation Modelling Practice and Theory 2016, 62, 68–87.
[36] ur Rehman, M.H.; Chee, S.L.;Wah, T.Y.; Iqbal, A.; Jayaraman, P.P. Opportunistic computation offloading in mobile edge cloud computing environments. In Proceedings of the 2016 17th IEEE International Conference on Mobile Data Management (MDM). IEEE, 2016, Vol. 1, pp. 208–213.
Point 12: In Section 4.1, the authors claim that the formulated problem is an NP-hard problem. However, I cannot find a direct evidence/method to prove that problem is an NP-hard problem. A clear proof by reduction is required.
Response 12:
Please forgive me for my carelessness. I should add the citation to this point here. Chen et al. have studied the computation offloading problem in multi-user MEC environments. They proved that it is NP-hard to obtain a centralized optimal solution, and propose a game theoretic approach to achieve optimal offloading decisions in a distributed manner in their paper “Efficient Multi-User Computation Offloading for Mobile-Edge Cloud Computing” which is published in IEEE/ACM Trans. Networking, vol. 24, no. 5, Oct. 2016. Besides, another researcher also hold this view. Such as Mehrabi(Device-enhanced MEC: Multi-access edge computing (MEC) aided by end device computation and caching: A survey, IEEE Access, 2019),Mingfang Huang(A cloud--MEC collaborative task offloading scheme with service orchestration, IEEE Internet of Things Journal, 2019).I add these refernces into the manuscript.
Point 13: The authors should analyze the time complexity of the proposed algorithms in detail.
Response 13:
We add an analysis of the time complexity of the proposed algorithms in the conclusion section, illustrate the advantages of the them and the optimization direction in the future.
Point 14: At end of section 5, authors must compare their results to selected close papers from section 2.
Response 14:
We regret that our work let you down. Since the experimental data sets or source code in other works are unavailable, we decide to select the EUA and designed the existing experimental scheme. In fact, RS, NPPS and NPSBPP are abstrated from many common experiment scenarios . In order to enhance the persuasiveness of the experimental results, we constructed 5 groups of user randomly. Besides, like other papers, we take the execution efficiency and network bandwidth as performance indicators to evaluate the scheduling strategy.
The attached file is our updated manuscript, and the blue parts are the revised contents according to your comments.

Reviewer 2 Report
The authors propose a mobility-aware tasks assignment strategy to deal with tasks offloading for mobile edge computing. I found a very careless written document. There are duplicated paragraphs, typos, acronysms that need expansion and inconsistencies. The manuscript needs a hard work to improve readability.
For instance, related works section is out of focus. It doesn't provide a clear view of efforts in line with the main topic of the paper which is dealing with mobility for task offloading. It is purely descriptive and lacks knowledge abstraction in the topic. Please, provide a table that compares this work with other related works in the topic including [6,7,8]. For introducing challenges related to resource allocation in MUCEN, consider surveys such as [5].
Moreover, I found many inappropriatte references in the document, for instance:
In line 100: references 32 and 33. More specific literature has been written in mobile edge computing task offloading topic, e.g.[1,2,3,4]
Other example: reference 43 in line 156. which is the relation with tasks assignment of edge servers?
Problem statement needs clarification. Which is the user device role? It consumes or provides resources for tasks execution?
In experiment section, is dataset open for verification or enabling other research? If not, a descriptive section of the dataset is necessary to know more about data contained.
Which is the reasoning of having edge_cloud bandwidth higher than end_to_edge bandwidth? How about latency in both cases?
Section 5.3.1, could authors specify the criteria followed for grouping users?
Table 2, tasks cycle requirements are missing. How many sub-tasks compose a task?
In figure 5.a, How number running tasks is determined? Comparison with RS seems unfair since RS does not include tasks paralelization.
[1] Mach, Pavel, and Zdenek Becvar. "Mobile edge computing: A survey on architecture and computation offloading." IEEE communications surveys & tutorials 19.3 (2017): 1628-1656.
[2] Shakarami, Ali, Mostafa Ghobaei-Arani, and Ali Shahidinejad. "A survey on the computation offloading approaches in mobile edge computing: A machine learning-based perspective." Computer Networks 182 (2020): 107496.
[3] Kumar, Karthik, et al. "A survey of computation offloading for mobile systems." Mobile networks and Applications 18.1 (2013): 129-140.
[4] Wang, Jianyu, et al. "Edge cloud offloading algorithms: Issues, methods, and perspectives." ACM Computing Surveys (CSUR) 52.1 (2019): 1-23.
[5] Hirsch, M., Mateos, C., & Zunino, A. (2018). Augmenting computing capabilities at the edge by jointly exploiting mobile devices: A survey. Future Generation Computer Systems, 88, 644-662.
[6] Mukherjee, A., & De, D. (2016). Femtolet: A novel fifth generation network device for green mobile cloud computing. Simulation Modelling Practice and Theory, 62, 68-87
[7] Chen, X., Pu, L., Gao, L., Wu, W., & Wu, D. (2017). Exploiting massive D2D collaboration for energy-efficient mobile edge computing. IEEE Wireless communications, 24(4), 64-71.
[8] ur Rehman, M. H., Chee, S. L., Wah, T. Y., Iqbal, A., & Jayaraman, P. P. (2016, June). Opportunistic computation offloading in mobile edge cloud computing environments. In 2016 17th IEEE International Conference on Mobile Data Management (MDM) (Vol. 1, pp. 208-213). IEEE.
Author Response
Dear reviewer:
Thank you for your decision and constructive comments on my manuscript. We have carefully considered the suggestion of Reviewer and make some changes. We have tried our best to improve and made some changes in the manuscript. Please find my itemized responses in below and my revisions/corrections in the re-submitted files. Revision notes, point-to-point, are given as follows:
Point 1:
For instance, related works section is out of focus. It doesn't provide a clear view of efforts in line with the main topic of the paper which is dealing with mobility for task offloading. It is purely descriptive and lacks knowledge abstraction in the topic. Please, provide a table that compares this work with other related works in the topic including [6,7,8]. For introducing challenges related to resource allocation in MUCEN, consider surveys such as [5].
Response 1:
Thank you very much for the recommended literatures. I have carefully read all the aritcles and some references cited in them. I reorganize the “Related works” of the manuscript, delete some literatures with weakly relevant content, and update several references including all of the recommended literature.
In section 2, we summarize traditional task offlaoding methods in the first two paragraphs, and analyze limitations of this type of method when the edge resources are insufficient in paragraph 3. Paragraph 4 introduces several task offloading approaches with prediction of user trajectory, meanwhile, we illustrate the limitations of these methods and the reasons for the limitations. At the end of this section, we briefly describe our method and how our method overcome the weaknesses of other approaches.
Point 2: Moreover, I found many inappropriatte references in the document, for instance:
In line 100: references 32 and 33. More specific literature has been written in mobile edge computing task offloading topic, e.g.[1,2,3,4]
Other example: reference 43 in line 156. which is the relation with tasks assignment of edge servers?
Response 2:
Please forgive our mistakes. I have corrected the inappropriate references you pointed, deleted some literatures with weakly relevant content, and replaced them by another papers listed in the following:
[2] Shakarami, A.; Ghobaei-Arani, M.; Shahidinejad, A. A survey on the computation offloading approaches in mobile edge computing: A machine learning-based perspective. Computer Networks 2020, 182, 107496.
[26] Kumar, K.; Liu, J.; Lu, Y.H.; Bhargava, B. A survey of computation offloading for mobile systems. Mobile networks and Applications 2013, 18, 129–140.
[27] Chen, X.; Pu, L.; Gao, L.;Wu,W.;Wu, D. Exploiting massive D2D collaboration for energy-efficient mobile edge computing. IEEE Wireless communications 2017, 24, 64–71.
[31] Hirsch, M.; Mateos, C.; Zunino, A. Augmenting computing capabilities at the edge by jointly exploiting mobile devices: A survey.Future Generation Computer Systems 2018, 88, 644–662.
[32] Mukherjee, A.; De, D. Femtolet: A novel fifth generation network device for green mobile cloud computing. Simulation Modelling Practice and Theory 2016, 62, 68–87.
[36] ur Rehman, M.H.; Chee, S.L.;Wah, T.Y.; Iqbal, A.; Jayaraman, P.P. Opportunistic computation offloading in mobile edge cloud computing environments. In Proceedings of the 2016 17th IEEE International Conference on Mobile Data Management (MDM). IEEE, 2016, Vol. 1, pp. 208–213.
Point 3: Problem statement needs clarification. Which is the user device role? It consumes or provides resources for tasks execution?
Response 3:
User device is the center of User-centered edge mobile network. I have updated the illustration of the User device on the page 5.
Point 4: In experiment section, is dataset open for verification or enabling other research? If not, a descriptive section of the dataset is necessary to know more about data contained.
Response 4:
EUA is an public data set, there are numerous studies use this dataset as the basis for their experiments. I have updated the details about the data set in the experiment, and add the link address of EUA on page 10.
Point 5: Which is the reasoning of having edge_cloud bandwidth higher than end_to_edge bandwidth? How about latency in both cases?
Response 5:
In tranditional computer networks, there is a gap among different level networks. I just assume that there is a difference in performance between end-edge and edge-cloud in our experiments.
Point 6: Section 5.3.1, could authors specify the criteria followed for grouping users?
Response 6:
We constructed 5 groups randomly from the 816 users. The detailed content is described on page 10. Moreover, the table 3 contains the parameters declaration.
Point 7: Table 2, tasks cycle requirements are missing. How many sub-tasks compose a task?
Response 7:
To complete the parallel experiment, we assume that each task contains 2 to 5 subtask.
Point 8: In figure 5.a, How number running tasks is determined? Comparison with RS seems unfair since RS does not include tasks paralelization.
Response 8:
The number of running tasks is derived from the number of service requests at a certain time. The service request is generated randomly, and we have saved the stochastic data in some files for reproducing the experimental results. RS is the baseline for the another three experimental scenarios. A task’s execution time is the sum of the running time of all of the subtasks.
The attached file is our updated manuscript, and the blue parts are the revised contents according to your comments.

Reviewer 3 Report
Please see attached file for comments.

Author Response
Dear reviewer:
Please see attachment.

Reviewer 4 Report
This work discussed a mobility-aware parallelized task offloading for mobile edge networks. Several relevant contributions are well presented. However, there are several comments addressed to the authors
1. There are grammatical issues. Particular missing space before introducing references number. The consistency in formating the whole manuscript is required
2. My greatest concern is about the architecture of User-centered edge mobile network. This architecture is different from the traditionnal MEC. It introduces mobility presiction and dynamic configuration of the topology. How these claimed features of this architectures are evaluated and compared to MEC architecture.
3. The evaluation metrics of this user-centered architecture must be presented, discussing the main benchmark and advantages compared to MEC
4. Future directions and issues regarding mobile user-centered architectures might bring new research opportunities to researcher community. How about adding a short paragraph in discussion section.
Author Response
Dear reviewer:
Thank you for your decision and constructive comments on my manuscript. We have carefully considered the suggestion of Reviewer and make some changes. We have tried our best to improve and made some changes in the manuscript. Please find my itemized responses in below and my revisions/corrections in the re-submitted files. Revision notes, point-to-point, are given as follows:
Point 1:
There are grammatical issues. Particular missing space before introducing references number. The consistency in formating the whole manuscript is required.
Response 1:
Thanks for your advice and reminder. I check the article again carefully and find some expression errors. I have modified the abstract, introduction, and related works etc. I aslo revised some errors in grammar and presentation. For example:
Line 61:” constructed” -> ”construct”
Line 62: ” surmised” -> ” surmise”
Line 65: “…adapt to the topology changes of UCENs based on the user's movement.” -> “adapt to the topology changes of UCENs.”
Line 65: “distributed” -> “distributes”
Line 67: “showed” -> “show”
Line 69: “analyzed” -> “analyze”
Line 75: “… current location and moving direction” -> “… current location and the moving direction”
Line 78: “UEs can build…” -> “UEs construct…”
Line 81: “…the computing devices…” -> “…the available computing devices…”
Line 83: “A offloading strategy …” -> “An offloading strategy …”
Line 158: “we shall first formulate the mobile” -> “we formulate the mobile”
Line 166: “rebuilding of a new topology” -> “rebuilding of a new network topology”
Line 172: “, andMis the number…” -> “where M is the number…”
Line 174: “S is the number…” -> “where S is the number…”
Line 203: “…offloading algorithm in this paper.” -> “…d offloading algorithm paper.”
Line 230: “, edge and end.” -> “, edge and end respectively.”
Line 280: “moving UEs is a irregular.” -> “moving UEs is irregular.”
In addition, I corrected some punctuation errors directly in the original draft.
Point 2: My greatest concern is about the architecture of User-centered edge mobile network. This architecture is different from the traditionnal MEC. It introduces mobility presiction and dynamic configuration of the topology. How these claimed features of this architectures are evaluated and compared to MEC architecture.
Response 2:
The components of user-centered edge mobile network is the same as traditional MEC which is an edge-centered network. The resource in traditional MEC is managed by the edge server, it provides a reactive service response model, once the resources managed by the edge is inadequate, the center will request the assistance of cloud or other device. While, the user-centered edge mobile network is dynamic, the moving terminal device and the edge server jointly manage the resources on devices and edge servers. When the edge server make resource distribution scheme, it needs to consider the mobile status of the user.
Point 3: The evaluation metrics of this user-centered architecture must be presented, discussing the main benchmark and advantages compared to MEC
Response 3:
User-centered architecture is a temporary network form during the user’s moving. It has obvious temporal feature. It may provide a personalized list of available devices for any end user. The task offloading policy in such a network architecture assures that the solution is as rational as feasible if user displacement is taken into consideration.
Point 4: Future directions and issues regarding mobile user-centered architectures might bring new research opportunities to researcher community. How about adding a short paragraph in discussion section.
Response 4:
Thanks for you suggestion. I add a paragraph to disscuss the mobile user-centered architectures at the end of the manuscript. Since the user device participates in managing the resource of edge networks, we select the security of the network as a potential area for research in our next works.
The attached file is our updated manuscript, the yellow parts are revised contents according to your comments.

Round 2
Reviewer 1 Report
Since the previous version, the authors have done huge work, and the paper is much better. This version looks good. Therefore, I suggest accepting this paper after handling these comments:
· The abstract needs to summarize the performance evaluation results concerning the benchmark approaches using the percentage improvement.
· More details about the simulation software exploited still need to be added.
· In equation 11, the authors still need to clarify with more detail how the time and energy are added together, however, each of them has a different unit.
· The authors mentioned that RS, NPPS, and NPSBPP are abstracted from many common experiment scenarios. They should specify the papers that applied these scenarios and compare with them.
· I recommend the authors to read and cite the following paper, which is recently done and will be helpful to the revision of this paper:
1. "A Mobility-Aware and Fault-Tolerant Service Offloading Method in Mobile Edge Computing.", 2022.
2. " Mobility-aware multi-hop task offloading for autonomous driving in vehicular edge computing and networks.", 2022.
3. "V2V Task Offloading Algorithm with LSTM-based Spatiotemporal Trajectory Prediction Model in SVCNs.", 2022.
4. " Secure Task Offloading in Blockchain-Enabled Mobile Edge Computing with Deep Reinforcement Learning.", 2022.
Author Response
Dear reviewer:
Thanks very much for taking your time to review this manuscript. I have revised the manuscript. The following is the replies to your comments.
Point 1:
The abstract needs to summarize the performance evaluation results concerning the benchmark approaches using the percentage improvement.
Response 1:
Thanks for your advice and reminder. I compare the task offloading hit rate and bandwidth occupancy to the benchmark approaches by using percentage improvement.
Point 2: More details about the simulation software exploited still need to be added.
Response 2:
Please forgive us for the careless. We have add the description of the software-tools and programming language used in our experiments. Additionally, the experimental data set has illustrated in the same paragraph.
Point 3: In equation 11, the authors still need to clarify with more detail how the time and energy are added together, however, each of them has a different unit.
Response 3:
The illustration of equation 11 are represented in the middle of page 9 and the table 3 on page 13.
Point 4: The authors mentioned that RS, NPPS, and NPSBPP are abstracted from many common experiment scenarios. They should specify the papers that applied these scenarios and compare with them.
I recommend the authors to read and cite the following paper, which is recently done and will
be helpful to the revision of this paper:…
Response 4:
Thanks for your comments. Firstly, we can not find a similar essay to RS, NPPS, and NPSBPP in our references. For example, most position predection methods[2,14 ,19,21,24] is based on a priori data set. Since the sensitivity of users data, we could not obtain these contrast datasets. Meanwhile, it is difficult to reproduce the experiments in these comparing papers. To maintain the experimental rigour of the article, we have added a description of the experimental scenario.
Additionally, I have carefully read your recommended papers which are useful for us to revise our manuscript. We also cite some of them in the paper.
With best wishes,
Yours sincerely,
Pu Han

Reviewer 2 Report
Major issue:
Table 1 needs further elaboration. Having a single row makes no sense. The main focus is still offloading and not mobility-aware as I pointed out in the first revision round. Could authors put some extra effort into identifying some mobility-aware related categories?
minor issues.
Problems described in Reference 10 share commonalities with Edge Computing but are too specific to WSN. Please, change it.
The manuscript is full of typos, some of them are in
line 109: thenetworks
line 119: Responeded? novle?
line 125: "it can guarantee that the offload policy is bit optimum at a given moment." This phrase hasn't the rigor of a scientific study, please fix it.
Table 3: Please elaborate on the reasoning for selecting 5 group users for each scenario.
"Response 7: To complete the parallel experiment, we assume that each task contains 2 to 5 subtask." Has the revised version this clarification? I don't see it.
Finally, if authors saved data generated stochastically in files for facilitating experiments reproduction, Could authors make them publicly available in a GitHub repository for example?
Author Response
Dear reviewer:
Thanks very much for taking your time to review this manuscript. I have revised the manuscript. The following is the replies to your comments.
Point 1:
Major issue:
Table 1 needs further elaboration. Having a single row makes no sense. The main focus is still offloading and not mobility-aware as I pointed out in the first revision round. Could authors put some extra effort into identifying some mobility-aware related categories?
Response 1:
I am so sorry for my carelessness. In this version, I have revised the table 1, and modifid the description on page 3. The further elaboration of the task offlaoding methods is presented in table 1, we also analyzed the shortcomings of each type of the approach.
Point 2:
minor issues.
Problems described in Reference 10 share commonalities with Edge Computing but are too specific to WSN. Please, change it.
Response :
Thanks for your reminder. I replaced this reference with other one.
The manuscript is full of typos, some of them are in
line 109: thenetworks
line 119: Responeded? novle?
Response :
Thanks for your advice and reminder. I recheck the article carefully, and find another similar errors. Such as:
Line 112: “thenetworks” ---> ” the networks”
Line 121: “novle” ---> ” novel”
Line 291: “algorthms”---> ” algorithms”
line 125: "it can guarantee that the offload policy is bit optimum at a given moment." This phrase hasn't the rigor of a scientific study, please fix it.
Response :
This part has been updated in the current version.
Table 3: Please elaborate on the reasoning for selecting 5 group users for each scenario.
Response :
The 5 groups users are randomly selected from the 816 users in EUA dataset. In addition, the randomness of the users’ behavior are maintained by setting various speed, task requirements to each user.
"Response 7: To complete the parallel experiment, we assume that each task contains 2 to 5 subtask." Has the revised version this clarification? I don't see it.
Response :
Please forgive my mistake. I just reply your comments in the last responding file, and forget to add these descriptions to our manuscript. In current version, I update the content in the table 3.
Finally, if authors saved data generated stochastically in files for facilitating experiments reproduction, Could authors make them publicly available in a GitHub repository for example?
Response :
Thanks for for your interest in our work. We think your suggestion is a good idea. Next, we need to organize the work of this article and upload it to GitHub.
With best wishes,
Yours sincerely,
Pu Han

Reviewer 3 Report
This is a better version of the paper, but still has some issues.
You did not consider some of my previous comments, and I address some of them again. See attached file.
Section 2 is the weakest section of the paper.
The results section could be more detailed.

Author Response
Dear reviewer:
Thanks very much for taking your time to review this manuscript. I have revised the manuscript. The following is the replies to your comments.
Point 1:
Typographical errors marked in the manuscript.
Response 1:
Thanks for your reminder. I have corrected all of mistakes marked by you in the manuscript. Additionally, I find some another similar errors and correct them. Such as:
Line 112: “thenetworks” ---> ” the networks”
Line 121: “novle” ---> ” novel”
Line 291: “algorthms”---> ” algorithms”
Point 2: Despite being better, this section is still lacking a proper analysis of works. I past my previous comment: "This section is an enumeration of works. There is not a proper discussion on the authors contributions, results, shortcomings and strengths. How does this paper compares to the remaining? Does it solve the gaps of previous works? If it does not solve them, does it perform any better? I cannot see any discussion regarding these ideas which are crucial in a section such as this."
Response 2:
I am so sorry for my carelessness. In this version, I modifed the table 1 which summarises the methods of task offloading in categories and analyses the disadvantages of the different approaches. The comparision, marked with highlight, is represneted in paragraph 2 and 3 of this section.
With best wishes,
Yours sincerely,
Pu Han
Reviewer 4 Report
The authors have addressed all my comments.
Author Response
Dear reviewer:
Thanks very much for taking your time to review this manuscript. I have recheck my manuscript, and done some revising. The attachment is the modified version.
With best wishes,
Yours sincerely,
Pu Han

Round 3
Reviewer 2 Report
The authors have addressed all my questions. Thank you.